# Three-Dimensional Instance Segmentation Using the Generalized Hough Transform and the Adaptive n-Shifted Shuffle Attention

**DOI:** 10.3390/s24227215

**Published:** 2024-11-12

**Authors:** Desire Burume Mulindwa, Shengzhi Du, Qingxue Liu

**Affiliations:** 1Department of Electrical Engineering, Tshwane University of Technology, Pretoria 0001, South Africa; dus@tut.ac.za; 2School of Mechanical and Electrical Engineering, Kunming University, Kunming 650214, China; qingxueliu@gmail.com

**Keywords:** activation functions, attention mechanisms, generalized Hough transform, 3D instance segmentation

## Abstract

The progress of 3D instance segmentation techniques has made it essential for several applications, such as augmented reality, autonomous driving, and robotics. Traditional methods usually have challenges with complex indoor scenes made of multiple objects with different occlusions and orientations. In this work, the authors present an innovative model that integrates a new adaptive n-shifted shuffle (ANSS) attention mechanism with the Generalized Hough Transform (GHT) for robust 3D instance segmentation of indoor scenes. The proposed technique leverages the n-shifted sigmoid activation function, which improves the adaptive shuffle attention mechanism, permitting the network to dynamically focus on relevant features across various regions. A learnable shuffling pattern is produced through the proposed ANSS attention mechanism to spatially rearrange the relevant features, thus augmenting the model’s ability to capture the object boundaries and their fine-grained details. The integration of GHT furnishes a vigorous framework to localize and detect objects in the 3D space, even when heavy noise and partial occlusions are present. The authors evaluate the proposed method on the challenging Stanford 3D Indoor Spaces Dataset (S3DIS), where it establishes its superiority over existing methods. The proposed approach achieves state-of-the-art performance in both mean Intersection over Union (IoU) and overall accuracy, showcasing its potential for practical deployment in real-world scenarios. These results illustrate that the integration of the ANSS and the GHT yields a robust solution for 3D instance segmentation tasks.

## 1. Introduction

In computer vision, 3D instance segmentation has become a critical component, especially in applications such as autonomous vehicles, robotics, and virtual reality [1]. Compared with 2D segmentation, 3D segmentation offers a more complete understanding of a scene because 3D data have richer geometric scale and shape data with less background noise [2]. Within indoor environments, the ability to precisely detect and delineate individual objects in a 3D scene is very important because it helps to recognize and interact with the objects present [3]. Nevertheless, there are challenges posed by the different object sizes, scales, occlusions, orientations, and so on in the point cloud data. The most important problem in 3D instance segmentation is the capacity to extract and represent various features from point clouds. Point clouds are known to be irregular and unordered [4]. This sets a major challenge for conventional deep learning architectures because of their reliance on grid-like data structures. Lately, researchers have introduced point-based networks (PointNet [5] and PointNet++ [6]), which not only directly operate on point clouds but also overcome the limitations of voxelization. However, these methods still battle to handle complex scenes with multiple overlapping objects and capture long-range dependencies.

In this work, the authors propose a novel approach that combines the strength of the adaptive n-shifted shuffle (ANSS) attention mechanism and the Generalized Hough Transform (GHT) for 3D instance segmentation. The proposed ANSS uses an n-shifted sigmoid activation function that dynamically adjusts the focus of the network on relevant regions of the point cloud. This mechanism enhances the model’s ability to distinguish between objects with similar geometric features by adaptively reshuffling the feature space, resulting in more accurate segmentation. Moreover, the n-sigmoid activation function also helps mitigate the vanishing gradient problem by maintaining a more stable gradient flow during backpropagation. This further improves the learning process, particularly in deeper layers, leading to better performance in complex 3D segmentation tasks.

The proposed n-shifted sigmoid activation function plays a crucial role in the attention mechanism. It is, in fact, an evolution from the n-sigmoid activation function, which provides a smooth and differentiable tool to oversee the attention weights [7] but here is in the range [−1, 1]. By shifting the sigmoid function, the authors allow the network to fine-tune its focus on the feature space’s various parts, so the overall performance of the attention mechanism is enhanced.

The incorporation of GHT [8] further strengthens this model with the provision of a vigorous structure that localizes and detects various objects within the 3D environment. GHT is remarkably effective when it comes to object orientation and partial occlusions, which makes it essential for handling challenging segmentation tasks [8].

The authors evaluate the proposed model on the Stanford 3D Indoor Spaces Dataset (S3DIS), a widely used benchmark dataset for 3D instance segmentation. The experiments prove that the proposed approach outperforms state-of-the-art methods in both mean Intersection over Union (IoU) and overall accuracy. The combination of ANSS and GHT results in a model that is not only more accurate but also more robust to the challenges posed by 3D instance segmentation.

In summary, the contributions of this work are that it achieves the following:Introduced an innovative attention mechanism, the ANSS attention, which improves the model’s capacity to concentrate on relevant parts of the 3D space, thus capturing subtle details better than conventional approaches.Reformulated and improved the n-sigmoid activation function to enable the model to output values in the range of [−1, 1], which allows the representation of negative relationships between features and actively suppresses conflicting features (negative values near −1).Reformulated the Generalized Hough Transform with deep learning integration and the addition of a new attention mechanism for 3D instance segmentation.Improved the 3D object detection performance on 3DSIS, a benchmark dataset.

The rest of this paper is organized as follows: Section 2 reviews the related work on 3D instance segmentation, focusing, among others, on attention mechanisms and Hough Transform-based methods. Section 3 details the architecture of the proposed model, including the ANSS attention module and its integration with GHT. Section 4 presents the experiments and results, where a discussion on the potential future research directions is included, and Section 5 concludes this paper.

## 2. Related Works

### 2.1. Point-Based Networks

Three-dimensional instance segmentation has greatly evolved in recent years, with an increasing number of techniques that aim to tackle the difficulties posed by 3D point cloud data [9,10,11]. Early approaches relied heavily on voxelization, converting point clouds into a structured grid format that could be processed using 3D convolutional neural networks (3D CNNs) [12]. Models such as VoxelNet [13] and MinkowskiNet [14] have laid the groundwork for 3D segmentation by leveraging the power of CNNs to extract features from volumetric data. But these methods suffer from high computational costs and memory ineptitudes because of the sparsity of point cloud data, leading to the point-based network’s expansion.

PointNet and its successor, PointNet++, marked a significant shift in point cloud processing. The innovation of PointNet was to directly operate on raw point clouds by means of a series of multi-layer perceptrons (MLPs) to learn point-wise features, followed by a max-pooling operation to aggregate global features. PointNet++ went further and integrated hierarchical feature learning, allowing the network to capture local structures at multiple scales. These methods demonstrated the potential of directly processing point clouds, but they also highlighted the limitations of MLP-based architectures in capturing complex spatial relationships.

### 2.2. Attention Mechanisms

Because of the limitations of the PointNet-based models, researchers explored attention mechanisms [15]. Attention mechanisms have become central to current deep learning models, especially in natural language processing and computer vision [16]. The ability to dynamically focus on relevant parts of the input data has proven instrumental for tasks that require fine-grained understanding [17]. In the context of 3D instance segmentation, attention mechanisms have been used to enhance the model’s capability to differentiate between objects with comparable geometric features. Methods such as the Point Attention Network (PAN) [18] and Point Transformer [19] have incorporated attention layers to better capture long-range dependencies and context within point clouds.

Shuffle attention [20] is an advanced variant of the traditional attention mechanisms that enhances the model performance by rearranging feature maps before applying attention. The idea is to introduce a shuffling operation that diversifies how the model interprets the input data, encouraging it to look at features in new, varied ways. This reshuffling enhances the model’s discriminative power, making it more effective at learning and distinguishing complex patterns within the data.

### 2.3. Generalized Hough Transform

This is another critical component of the proposed model. GHT is a well-established technique for detecting shapes and patterns in images, traditionally used in 2D computer vision tasks. Its extension to 3D has been explored in various works [21,22,23,24], particularly for object detection and pose estimation [25,26,27]. The strength of GHT lies in its robustness to variations in object orientation and partial occlusions, making it an ideal choice for challenging 3D instance segmentation tasks. Recent works have integrated GHT with deep learning models to improve their ability to localize objects within 3D scenes [28,29]. However, these approaches often rely on handcrafted features or are limited to specific object categories.

### 2.4. Activation Functions

Activation functions are critical components of deep learning models, as they introduce non-linearity and help networks learn complex patterns [30]. Traditional activation functions, such as sigmoid, ReLU (Rectified Linear Unit) [31], ELU [32], and Tanh [33], have been widely used across various deep learning tasks. Sigmoid and Tanh are bounded functions, with sigmoid mapping values to the range [0, 1] and Tanh to [−1, 1]. These bounded ranges, however, have led to issues such as vanishing gradients, especially in deep networks. ReLU, on the other hand, introduced an unbounded, piecewise linear function, allowing faster convergence in many cases, but also introduced issues like dead neurons when the activation remains zero for negative inputs.

## 3. The Proposed Adaptive n-Shifted Shuffle (ANSS) Attention Integrated with the Generalized Hough Transform (GHT)

This proposed model combines the innovative ANSS and the GHT (Figure 1). From an input point cloud, the network (Figure 1 and Figure 2) predicts the object instance masks by leveraging an innovative ANSS to improve the 3D GHT. The full model consists of two main parts: the ANSS attention mechanism and the GHT with the PointNet backbone where objects in the 3D scene are instantiated. FCs are fully connected layers.

The adaptive n-shifted shuffle (ANSS) mechanism combines three components (adaptive shuffle (AS), n-shifted sigmoid activation function (n*ρ*), and shuffle attention SA) (Figure 2) to prioritize and rearrange feature focus across spatial regions to improve segmentation in complex 3D scenes. The GHT implements robust 3D object detection and localization using PointNet and a GHT module.

Table 1 below showcases the sizes and detailed layers of the proposed model. The GHT here does not explicitly perform a voting process like traditional Hough Transform approaches [34,35,36,37].

### 3.1. Adaptive n-Shifted Shuffle (ANSS)

The ANSS comprises three components, which are the n*ρ* (n-shifted sigmoid), AS (adaptive shuffle), and SA (shuffle attention).

#### 3.1.1. n-Shifted Sigmoid Activation Function

The n-shifted sigmoid activation function represents a pivotal innovation in the architecture of the ANSS GHT model, contributing to its effectiveness in complex tasks like segmentation and object detection.

a.n-sigmoid activation function

The n-Sigmoid *ρ* (Equation (1)) [7] is a variant of the standard sigmoid function, where the steepness of the curve is controlled by a factor n:(1)$$\rho (x,\;n)=\frac{\alpha }{1+e^{nx+\beta }}$$
where n is a constant that depends on *α* and results from the computation of the integration of the derivative, and *β* is the offset parameter that centers the input domain x, while *α* ∈ R is the logistic growth rate parameter.

This function allows more flexibility in controlling the slope of the activation curve. This added flexibility enables better control over how the model responds to different magnitudes of input values, which can be beneficial in scenarios where precise feature scaling is required.

b.n-shifted sigmoid activation function

The n-shifted sigmoid function n*ρ* in equation (Equation (2)) is an evolution from the n-sigmoid activation function that is designed to address its limitations by shifting the output range to [−1, 1]. Mathematically, it is expressed as
(2)$$\;\;n\rho \left(x,n\right)=\frac{2}{1+e^{\left(-nx+\beta \right)}}-1$$
where n is a constant that depends on *α* and results from the computation of the integration of the derivative, and *β* is the offset parameter that centers the input domain x, while *α* ∈ R is the logistic growth rate parameter.

Unlike the n-sigmoid, which outputs values between [0, 1], this shifted version is symmetrical around zero, producing values in the range [−1, 1]. This symmetry can improve gradient flow and model convergence by allowing both positive and negative responses to the input data.

c.Benefits of the n-shifted sigmoid function in attention mechanisms

In the context of attention mechanisms, the n-shifted sigmoid function offers several advantages, as follows:An enhanced representation of the negative relationships: In traditional attention mechanisms, the sigmoid function (Equation (3)) is in the range [0, 1], which restricts the model to positive relationships. However, with n-shifted sigmoid (Equation (4)), the range is [−1, 1], expressing both positive and negative relationships between features. This is particularly useful in occluded spaces where the presence of one feature may inhibit or contradict another.

The standard sigmoid function:(3)$$\rho (x)=\frac{1}{1+e^{-x}}\to \;\rho \left(x\right)\epsilon [0,1]$$

To shift this range to [−1, 1], the authors introduced the n parameter and rescaled the output:(4)$$n\rho \left(x,n\right)=\frac{2}{1+e^{\left(-nx+\beta \right)}}-1$$

With this adjustment, n*ρ*$\left(x,n\right)\;$ now outputs values in the range [−1, 1], allowing the model to recognize negative relationships where attention needs to be reduced or negated.

An improved attention weight: The n-shifted sigmoid broader range [−1, 1] allows for finer-grained attention weights, where values can represent both strong positive influence (near 1) and strong negative influence (near −1).

The weighted attention A is calculated by applying n*ρ*$\left(x,n\right)$:(5)$$A(f)=f\cdot n\rho \left(x,n\right)$$

Here, the following hold:
○When $n\rho \left(x,n\right)$ ≈ 1, f receives amplified attention.○When n*ρ*$\left(x,n\right)$ ≈ −1, f is actively suppressed, giving the model suppressive control over attention weights.

An enhanced model stability and training: The n-shifted sigmoid broader range can help improve gradient flow and stability during training by mitigating the vanishing gradient problem. With the traditional sigmoid, outputs are restricted to [0, 1], and as x → ±∞, the gradient *ρ*$\prime \left(x\right)$ → 0, causing gradients to vanish.With n-shifted sigmoid, this issue is reduced, mainly because the output range [−1, 1] allows for more substantial gradients, even for large inputs, and the shift parameter n can be adjusted during training, adapting the activation to changing input distributions.

An adaptability to complex tasks: Its ability to handle both positive and negative activations makes it particularly well suited for these tasks. This function enables the model to perform the following:
○Delineate Boundaries: Since the output can be negative, boundaries between contrasting regions can be clearly distinguished.○Identify Complex Patterns: The shift allows flexibility in the model’s attention distribution, helping it handle intricate and variable spatial arrangements.○Mathematical Intuition for Boundary Delineation: When a region with high contrast (boundary) is encountered, n-shifted sigmoid can amplify positive features on one side of the boundary while suppressing features on the other side.

For a feature map F with a boundary between regions R_1_ and R_2_, the attention weights (Equation (6)) from n*ρ*$\left(x,n\right)$ will enhance *F_R_*_1_ while attenuating *F_R_*_2_, yielding
(6)$$A(F)=[n\rho \left(F_{R1}\right)\;\approx \;1,\;n\rho \left(F_{R1}\right)\;\approx -1$$

This adaptability allows the model to effectively learn complex spatial relationships, leading to more accurate segmentation and classification in 3D scenarios.

#### 3.1.2. Adaptive Shuffle Pattern

The adaptive shuffle (AS) component is designed to dynamically rearrange or shuffle feature channels in response to the input’s spatial characteristics. This allows the network to create diverse feature representations without altering the spatial dimensions.

Assume the input feature map is *X* ϵ ℜ^*C*×*H*×*W*^, where *C* is the number of channels, and *H* and *W* are spatial dimensions.

After applying convolution:*X*′ = conv1 *×* 1(*X*) ϵ ℜ^*C*′×*H*×*W*^

Then, the feature map *X*′ is split into *G* groups:*X*′ = {X_1_, *X*_2_, …, *X*_G_}, where X_i_ ϵ ℜ*^C^*^′/*G*^^×*H*^^×*W*^

The channels within each X_i_ are then shuffled according to a learned or fixed shuffle pattern. This shuffle enhances spatial diversity by mixing features across different channels.

Once each group’s channels have been shuffled, the groups {*X*_1_, *X*_2_, …, *X_G_*} are concatenated back together along the channel dimension to form the final shuffled feature map. This new feature map retains the original spatial dimensions *H* and *W* but now has a rearranged channel structure that is more spatially diverse:X_shuffled_ = concat (X_1_, X_2_, …, X_G_) ∈ ℜ*^C^*^′^^×*H*^^×*W*^

This concatenation step completes the adaptive shuffle operation, yielding a transformed representation of the original input where the channels have been adaptively rearranged to better represent the spatial complexities of the data.

The adaptive shuffling approach provides several key advantages, such as enhanced spatial diversity (where the network can represent spatial details more effectively by capturing subtle variations and dependencies in the input data), more flexible representation learning (where the use of a learned shuffle pattern allows the model to adapt its channel rearrangements to specific features within the data), and stable spatial consistency (where it allows diverse feature recombination without losing structural integrity).

#### 3.1.3. Adaptive n-Shifted Shuffle Attention (ASA)

The proposed attention mechanism uses Global Average Pooling (GAP) to aggregate global information over the input features in its channel attention, just like the original shuffle attention (SA).

Instead of using two scaling and shifting parameters, the ANSS feeds the pooled features through a fully connected layer (Dense layer), followed by another dense layer to generate a channel-wise attention vector.
$$X{\mathrm{dense}}_{1}=W_{1}\cdot GAP\left(X\right)+b_{1},\;X{\mathrm{dense}}_{2}=W_{2}\cdot X{\mathrm{dense}}_{1}+b_{2},$$

After generating the adaptive attention vector, a custom n-shifted sigmoid activation function is applied to scale the output to a range of [−1, 1] rather than [0, 1]. Moreover, the adaptive shuffle attention module introduces a trainable shuffle pattern (the adaptive shuffle—Section 3.2.2), which is learned during training and allows the model to reorganize or reorder the channels dynamically, contrary to the fixed shuffling operator in the original shuffle attention.

The adaptive shuffle (AS) pattern is learned during training, allowing the model to reorder channels as follows:Split the feature map into groups: X′ = {X_1_, X_2_, …, X_G_}, where each X*_i_* ∈ ℜ*^C^*^/*G*^^×*H*^^×*W*^Shuffle the channels within each group based on a learned pattern.

This adaptive shuffling enables effective information flow across channels, enhancing feature interactions over varying orientations and scales. After reordering, the attention vector scales the input features elementwise:*X*_output_ = *X*⊙adaptive shuffle attention vector.

Unlike the original SA, the adaptive shuffle attention focuses on channel-wise attention and omits a separate spatial attention branch, concentrating on enhancing or suppressing specific features dynamically. The Adaptive Shuffle is designed to dynamically rearrange or shuffle feature channels in response to the input’s spatial characteristics. This allows the network to create diverse feature representations without altering the spatial dimensions.

### 3.2. Generalized Hough Transform (GHT)

#### 3.2.1. PointNet Backbone for 3D Point Cloud Processing

As the core model used for point cloud classification and segmentation tasks, PointNet (a groundbreaking approach in the field of 3D point cloud processing) operates directly on raw 3D point cloud data without needing voxelization or multi-view projections.

Traditional approaches to 3D data often involve converting point clouds into voxel grids or meshes, which can be computationally expensive and may lead to loss of information. PointNet addresses these challenges with a novel architecture specifically designed for unordered point sets by leveraging permutation invariance, point feature encoding, and transformation learning. The key innovation of PointNet is its ability to directly process raw point cloud data, bypassing the need for preprocessing steps like voxelization or meshing.

Since its introduction, PointNet has inspired several advancements and extensions, including PointNet++ (which improves local feature learning by incorporating hierarchical structures) [6] and various integrations with advanced neural network components.

#### 3.2.2. GHT Module

The GHT is traditionally a technique used in object detection to locate objects in images by mapping features to a reference point, such as an object’s center. In its classical form, GHT involves a “voting” mechanism where object features vote for potential positions of the object’s template, making it robust to variations in scale, rotation, and other transformations.

However, in this proposed model, the GHT is designed to work within a deep learning framework using 3D data.

a.GHT as a deep learning framework

The Generalized Hough Transform (GHT) has been revised for integration into a deep learning framework to generate the proposed Adaptive n-shifted sigmoid Generalized Hough Transform (ANSS GHT) approach. This reformulation involves modifications to make the process differentiable, scalable, and trainable within a neural network structure. Here is how:
-Differentiable Generalized Hough Transform: In its classic form, the GHT maps points in an image or 3D space to a parameter space and only relies on discrete operations that are not inherently differentiable, making backpropagation in a neural network impossible. So, the GHT can be modified to work with continuous probability density functions, so the vote accumulation in the Hough space becomes a continuous, smooth function. This is achieved by differentiable functions (SoftMax, sigmoid, n-shifted sigmoid, … for smoother accumulation of votes, enabling gradients to be propagated).-Adaptation of the “voting” process: Instead of hand-crafted features for mapping points to the Hough space, a neural network learns the optimal feature representations. Here, in the ANSS GHT, the n-shifted sigmoid function definitely allows scaling the voting intensities based on learned parameters, dynamically adapting to various spatial patterns and contexts in the 3D space.-Integration with convolutional and attention mechanisms: The reformulated GHT benefits from convolutional layers to extract hierarchical features from the 3D input. By using a convolutional backbone, the network can efficiently capture both local and global contexts.
b.GHT for object detection

Instead of relying on an explicit voting process, the GHT learns and aggregates local geometric information directly from the 3D point cloud data so that the model can infer object boundaries, centers, and other key features without the need for a traditional voting mechanism to generate instance masks.

By combining GHT with point features extracted from PointNet, the model can effectively perform object detection in 3D space. The GHT module processes these point features, learning to recognize object shapes and their spatial relationships. Convolutional layers and pooling operations are applied to capture fine-grained geometric details, enabling the model to locate objects even in cluttered or partially observed environments. This represents a pivotal innovation in the architecture of the ANSS GHT model, contributing to its effectiveness in complex tasks like segmentation and object detection.

## 4. Experiments and Results

To validate the effectiveness of the proposed 3D ANSS GHT instance segmentation method, the authors designed a comprehensive set of experiments.

### 4.1. Dataset and Evaluation Metrics

The dataset used for this research is the Stanford 3D Indoor Spaces Dataset (S3DIS) dataset [38], which consists of multiple 3D point clouds representing different indoor environments. It includes various object categories like chairs, tables, walls, and other furniture types, offering a challenging and realistic scenario for 3D instance segmentation.

In fact, the S3DIS dataset boasts six hefty indoor areas from three different buildings. It involves several annotated objects inside a variety of different rooms. Figure 3 represents the six areas of the dataset.

Various metrics, such as mean Intersection over Union (IoU), overall accuracy, robustness in noisy or occluded environments, and inference time, were used to evaluate the model’s performance.

#### 4.1.1. Data Preprocessing

The dataset was preprocessed to normalize the point clouds. Techniques used involved down sampling for computational efficiency, normalization of coordinates, and color augmentation. A grid subsampling technique was chosen to reduce the computational load while retaining important geometric details.

#### 4.1.2. Implementation Details

The model was trained on a GPU using the NVIDIA T4 set on a batch size of 32 and an initial learning rate of 0.001. A custom loss, called custom sparse SoftMax cross-entropy loss function, is implemented. Unlike standard SoftMax cross-entropy loss, which computes the loss across all possible classes, sparse SoftMax cross-entropy focuses on the sparse nature of the label data. This approach reduces the computational overhead and improves the model’s focus on relevant points, thereby enhancing segmentation accuracy.

A learning rate scheduler was implemented to reduce the learning rate as the training progressed. Adam optimizer with gradient clipping is used to prevent exploding gradients. To enhance the model’s generalization capabilities and improve its performance on unseen variations in point cloud data, data augmentation techniques (point cloud rotation, jittering, random flip) were employed.

### 4.2. Results and Analysis

#### 4.2.1. Average Recall (AR)

Mean average recall (mAR) is a metric mostly used to evaluate object detection and segmentation. It provides a measure of how well a model can detect relevant instances across different categories, averaged over all relevant objects. The mAR score is calculated by first finding the average recall (AR) for each object category and then averaging these AR scores across all categories. The proposed model achieved a mean AR of 71.61%, outperforming 3D BoNet (46.7%) and 3D-MPA (64.1%). The ANSS attention mechanism enabled the model to capture fine-grained details, particularly for small objects like chairs and monitors, which often pose a challenge in instance segmentation tasks.

Per-class instance segmentation results are shown in Table 2, which reports the mean average recall (mAR) scores when compared with other methods (3D-BoNet and 3D-MPA) on the various 13 individual classes. In most papers, the clutter class is not included, since it refers to miscellaneous objects that cannot fit into a specific and well-defined category. It also can vary greatly between different rooms and areas and introduces noise in the evaluation process.

In the S3DIS dataset, there are six areas. Area 5 and Area 6 are particularly interesting because they are often used as test sets in 3D segmentation tasks where they pose unique challenges. Area 5 contains a variety of room types, making it more challenging for models trained on other areas, while Area 6 plays an important role in contributing to the dataset’s overall diversity, containing its own set of unique indoor environments.

In Table 2, the best method and second-best method for each class are highlighted in bold font and underscore, respectively. In most cases, our method won and ranked second in other classes. The last line in Table 1 shows, when our method won, the improvement for each class achieved on the second-ranked method. Therefore, the proposed approach outperforms most state-of-the-art methods.

Table 3 reports scores on both Area 5 and six-fold cross-validation results. The metric is mean average recall (mAR) at an IoU threshold of 50%.

It is reported an increased recall of 14.57 mAR@50% (Area 5) and 7.5 mAR@50% (six-fold cross-validation) from the 3D-MPA and of 32.34 mAR@50% (Area 5) and 24.1 mAR@50% (six-fold cross-validation) from the 3D-BoNet, which means the proposed approach detects significantly more objects while simultaneously achieving higher precision.

#### 4.2.2. Accuracy

The mean Average Precision (mAP) is a popular metric used to evaluate the performance of object detection and instance segmentation. It combines both precision and recall to provide a balanced measure of a model’s accuracy and its ability to find all the relevant objects across multiple categories and Intersection over Union (IoU) thresholds.

In terms of accuracy, the proposed model achieves a mean Average Precision at 50% Intersection over Union (IoU), i.e., mAP@50% of 69.87%, compared with 66.7% with ASIS and 63.6% with 3D-MPA, for instance, on the 3DSIS six-fold CV. The superior accuracy is due to the combination of ANSS and GHT, allowing the model to better distinguish between overlapping objects, as seen in Table 4.

This paper reports scores on Area 5 and six-fold cross-validation results. The metric is mAP and mAR at an IoU threshold of 50%. The IoU is computed on per-point instance masks. This paper reports on the per-class mAP segmentation scores using the S3DIS dataset [34], as can be seen in Table 5. This paper reports the best results in nearly all categories. The conclusion is that this proposed method can detect significantly more objects with a higher precision.

The results of each class demonstrate that the n-shifted ANSS GHT is a powerful method in 3D instance segmentation.

#### 4.2.3. Robustness to Noise

Even after adding Gaussian noise to the point clouds, this proposed model maintained a high mAP@50 at 66.74%, showing a marginal degradation compared with its original non-noisy performance, which is 69.87%. Table 6 shows the comparison among the concerned methods on the degradation when noise is present. Our method is the most robust, since it displays the smallest degradation on mAP@50%. This suggests that the ANSS GHT architecture has strong robustness in handling noisy inputs because of the adaptive nature of the model’s attention mechanism, which dynamically shifts and adapts based on input features.

By focusing on important features and downplaying irrelevant noise, the model remains effective despite the added perturbations. The small decline in mAP@50 indicates that the architecture can successfully isolate valuable spatial information from noise, leading to minimal impact on its object detection and segmentation capabilities.

In contrast, models like 3D BoNet [39], which rely heavily on predefined feature extractors, or ASIS [42] and 3D-MPA [40], which may not dynamically adjust feature weights in response to noise, exhibit larger performance degradations (Table 6). Thus, the n-shifted shuffle attention mechanism provides a notable advantage in scenarios where point clouds are noisy or incomplete, reinforcing its utility in real-world 3D segmentation tasks.

#### 4.2.4. Inference Time

The Inference time is the time required for a trained machine learning model to process new, unseen data and generate a prediction. In 3D instance segmentation or object detection, inference time is measured from the moment the model receives an input (e.g., a 3D point cloud or an image) until it provides a prediction, such as labels or bounding boxes. The ANSS GHT method has an average inference time of 28.17 ms (Table 7). To benchmark the computation time with other state-of-the-art 3D instance segmentation methods and make a fair comparison, the authors ran the proposed framework on GPU using the NVIDIA T4. SGPN takes 170 ms, while 3D-MPA takes 300 ms. However, the detectionNet [41] takes 739 ms on an Nvidia Titan X GPU.

### 4.3. Ablation Studies

#### 4.3.1. Impact of n-Shifted Sigmoid

Experiments were conducted to investigate the impact of replacing the n-shifted sigmoid activation function with the standard ReLU [31] and other sigmoid functions (Table 8). The model with n-shifted sigmoid consistently outperformed other versions, achieving a mean AP improvement of 26.47% over ReLU, 15.7% over ELU [32], 28.57% over the original sigmoid activation function, and 6.52 over the n-sigmoid activation function. This shows that the smooth and adaptive behavior of the n-shifted sigmoid allows the model to refine attention weights more effectively.

#### 4.3.2. Impact of the ANSS Attention

To isolate the impact of the ANSS attention module, the authors first trained a version of the model without any attention mechanism, then with the normal shuffle attention, and finally with the inclusion of the ANSS. The mean AP dropped by 7.34% to 62.53%, confirming that the shuffle attention module plays a crucial role in enhancing feature discrimination, adaptive feature weighing, and model flexibility (Table 9).

#### 4.3.3. Impact of the Generalized Hough Transform (GHT)

Removing GHT from the architecture led to a 19.2% decrease in accuracy (mAP), particularly in scenes with significant occlusions or varied object orientations. Figure 4 shows that in the scene in Area_6 lounge_1, the occluded objects are not clearly demarcated, as opposed to when using the ANSS GHT (Figure 3). This demonstrates that GHT with an attention mechanism is essential for handling complex scenes with multiple objects and occlusions.

### 4.4. Visualization of Segmentation Results

#### 4.4.1. Qualitative Results

The authors provide visual examples of segmented scenes. Figure 5 shows the segmented scenes where the model successfully segments between objects, such as a table partially occluded by a chair.

#### 4.4.2. Failure Cases

The model occasionally struggled with objects that have very similar geometric structures but differ in texture or color, such as distinguishing between a sofa and a chair in certain cases. These failures highlight potential areas for further improvement (Figure 6).

### 4.5. Discussion

The results of this research indicate that the integration of the ANSS attention mechanisms and the GHT provides a powerful framework for 3D instance segmentation. The results of this research indicate that the integration of the ANSS attention mechanisms and the GHT provides a powerful framework for 3D instance segmentation.

The extensive experiments on the Stanford 3D Indoor Spaces Dataset (S3DIS) demonstrate that the proposed approach achieves state-of-the-art performance, with significant improvements in mean IoU, overall accuracy, robustness to noise, and inference time, and the impact of using the n-shifted sigmoid together with the adaptive shuffle attention has been tremendous. Additionally, the model exhibits strong resilience to noise and occlusion, maintaining high performance even under challenging conditions.

#### 4.5.1. Impact of the n-Shifted Sigmoid Activation Function

The authors evaluated the effect of replacing the n-shifted sigmoid activation function in the ANSS block with other activation functions. They considered four other activation functions—the original sigmoid, ReLU, ELU, and n-sigmoid—and compared them with the n-shifted sigmoid. Replacing the n-shifted sigmoid in the SE block with other well-known activation functions sizably reduced the model’s performance. This shows that the smooth and adaptive behavior of the n-shifted sigmoid does allow the model to refine attention weights more effectively.

#### 4.5.2. Impact of the ANSS Attention Module

To more effectively evaluate the impact of the ANSS attention module, the authors first trained a version of the model without the innovative attention mechanism and then with the normal shuffle attention without adding the innovative adaptive nature. Experiment results show that the shuffle attention module plays a crucial role in enhancing feature discrimination, adaptive feature weighing, and model flexibility because the model with the ANSS displayed excellent results.

#### 4.5.3. Impact of the Generalized Hough Transform (GHT)

The authors experimented with the complete removal of the GHT from the proposed architecture. This led to a sharp decrease in accuracy. Clearly, GHT with an attention mechanism is essential for handling complex scenes with multiple objects and occlusions.

#### 4.5.4. Future Research Directions

In the future, several avenues could be explored, such as the integration of additional geometric information into the attention mechanism. Another avenue would be ANSS GHT model optimization for real-time applications. The final possible avenue would be multi-modal approaches that combine point cloud data with other sensor inputs. By incorporating these features, the model will be able to better differentiate objects with similar shapes but different orientations or textures.

## 5. Conclusions

In this paper, the authors presented an innovative model that integrates a new adaptive n-shifted shuffle (ANSS) attention mechanism with the Generalized Hough Transform (GHT) for robust 3D instance segmentation of indoor scenes. The proposed technique uses the n-shifted sigmoid activation function, which improves the adaptive shuffle attention mechanism, permitting the network to dynamically focus on relevant features across various regions.

The integration of these components into a cohesive model enables it to outperform existing 3D instance segmentation methods, particularly in handling complex indoor scenes with multiple objects, noise, and occlusions.

Experiments conducted on the proposed method using the challenging Stanford 3D Indoor Spaces Dataset (S3DIS) proved its superiority over existing methods. The proposed approach achieves state-of-the-art performance in both mean Intersection over Union (IoU) and overall accuracy. These results illustrate that the integration of the ANSS and the GHT is a good answer for improved 3D instance segmentation tasks.

## Figures and Tables

**Figure 1 sensors-24-07215-f001:**
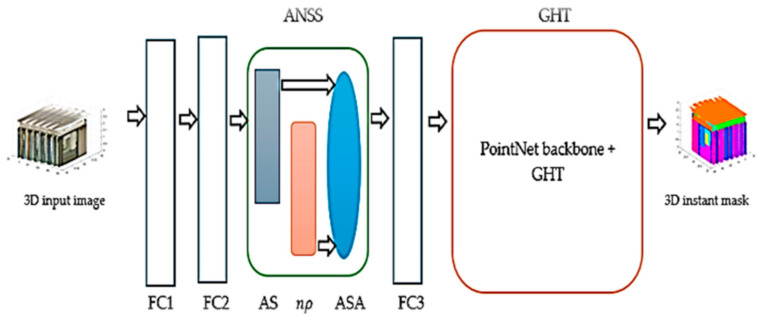
Three-dimensional n-shifted ANSS GHT network architecture. ANSS is made of the AS (adaptive shuffle) in orange, n*ρ* (n-sigmoid) in grey, and SA (adaptive shuffle attention) in light blue.

**Figure 2 sensors-24-07215-f002:**
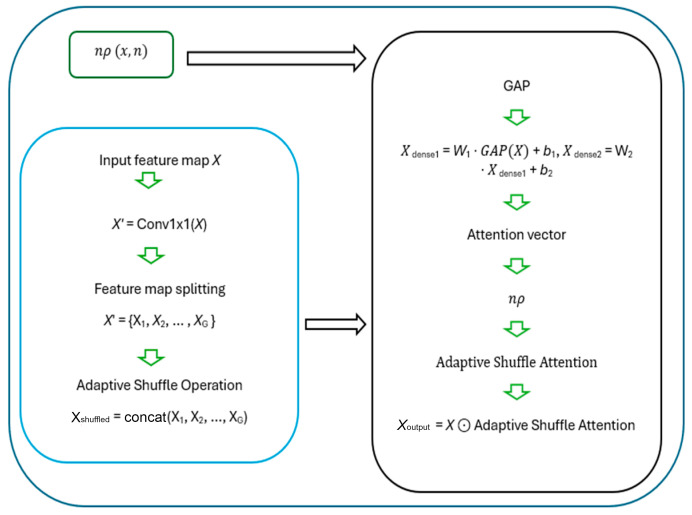
The detailed ANSS module of the proposed method.

**Figure 3 sensors-24-07215-f003:**
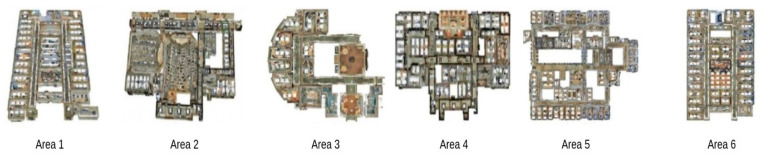
The six areas of the S3DIS dataset.

**Figure 4 sensors-24-07215-f004:**
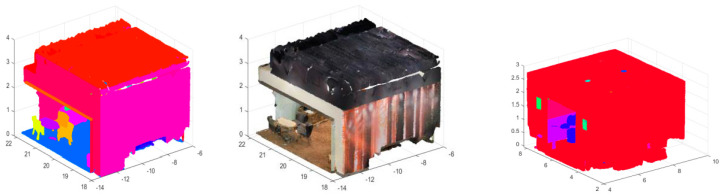
Three-dimensional instance segmentation from ANSS GHT (**left**), without GHT (**right**), and the original point cloud (**middle**). The model without GHT (**right**) found it difficult to partially segment all the chairs and even the wall and the curtain borders.

**Figure 5 sensors-24-07215-f005:**
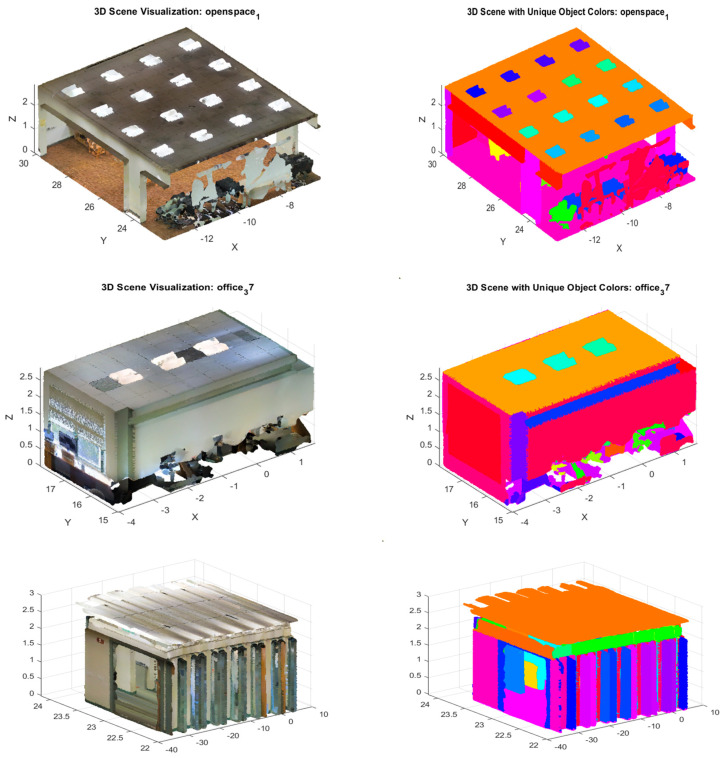
Visualization of the segmented scenes (**left**) in the S3DIS Dataset together with their 3D point clouds (**right**).

**Figure 6 sensors-24-07215-f006:**
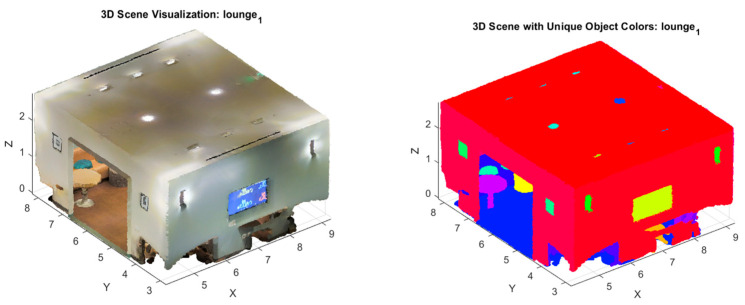
Example of a failure segmentation scene. Inside the door, the couches are not correctly segmented regarding the floor carpet.

**Table 1 sensors-24-07215-t001:** Detailed breakdown of the layer sizes and shapes of the ANSS GHT model.

Layer Name	Type	Input Shape	Output Shape	Parameters
FC1	Dense (128 units)	(batch_size, 4096, 3)	(batch_size, 4096, 128)	512
BN1	BatchNormalization	(batch_size, 4096, 128)	(batch_size, 4096, 128)	512
FC2	Dense (256 units)	(batch_size, 4096, 128)	(batch_size, 4096, 256)	33.024
BN2	BatchNormalization	(batch_size, 4096, 256)	(batch_size, 4096, 256)	1024
ANSS	Attention Layer	(batch_size, 4096, 256)	(batch_size, 4096, 256)	8576 (approx.)
FC3	Dense (num_classes)	(batch_size, 4096, 256)	(batch_size, 4096, num_classes)	Depends on num_classes
GHT	GHT Layer	(batch_size, 4096, 256)	(batch_size, 4096, num_classes)	10,240 (approx.)

**Table 2 sensors-24-07215-t002:** Per-class 3D instance segmentation scores on S3DIS. The bold data shows the best performing methods while the underlined ones indicate the second-best performing methods.

S3DIS 6-FoldCV	Ceiling	Floor	Wall	Beam	Column	Window	Door	Table	Chair	Sofa	Bookcase	Board	mAR
3D BoNet [39]	61.8	74.6	50.0	42.2	27.2	62.4	58.5	48.6	64.9	28.8	46.5	28.6	46.7
3D MPA [40]	68.4	96.2	51.9	**58.5**	**77.6**	**79.8**	69.5	32.8	75.2	**71.1**	**68.2**	38.2	64.1
SGPN [41]	58.42	83.6	42.2	25.6	7.15	42.73	45.2	38.25	47.0	0.00	13.5	31.68	31.68
Ours	**73.15**	**97.8**	**81.82**	56.5	71.1	68.7	**80.4**	**59.65**	**79.8**	61.75	64.65	**63.75**	**71.61**
Improvement	**4.75**	**1.6**	**29.08**	-	-	-	**10.9**	**11.05**	**4.6**	-	-	**22.55**	**7.5**

**Table 3 sensors-24-07215-t003:** Three-dimensional instance segmentation mAR@50 scores on S3DIS Dataset on mAR@50%.

Methods	S3DIS 6-Fold CV	S3DIS Area 5
3D-BoNet [39]	47.6	40.2
ASIS [42]	47.5	42.4
3D-MPA [40]	64.1	58.0
PartNet [43]	43.4	-
3D ANSS GHT (Ours)	**71.61**	**72.54**

**Table 4 sensors-24-07215-t004:** Three-dimensional instance segmentation scores on Area 5 and 6-fold cross-validation results on mAP@50%. The bold data shows the best performing methods while the underlined ones indicate the second-best performing methods.

Methods	S3DIS 6-Fold CV	S3DIS Area 5
3D-BoNet [39]	65.6	57.3
ASIS [42]	63.6	55.3
3D-MPA [40]	66.7	63.1
3D ANSS GHT (Ours)	**69.87**	**69.34**

**Table 5 sensors-24-07215-t005:** Per-class 3D instance segmentation mAP@50 scores on S3DIS.

S3DIS CV	Ceiling	Floor	Wall	Beam	Column	Window	Door	Table	Chair	Sofa	Bookcase	Board	mAP
3D BoNet [39]	88.5	89.9	64.9	42.3	48.0	**93.0**	66.8	55.4	72.0	49.7	58.3	**80.7**	65.66
3D MPA [40]	**95.5**	**99.5**	59.0	44.6	57.7	89.0	**78.7**	34.5	**83.6**	55.9	51.6	71.0	66.7
SGPN [41]	78.15	80.27	48.90	33.6	16.9	49.6	44.48	30.33	52.22	23.12	28.50	28.62	42.9
3D-BEVIS [44]	71.00	96.70	79.37	45.10	64.38	64.3	70.15	57.22	74.22	47.92	57.97	59.27	65.66
Ours	71.5	98.75	**80.25**	**55.6**	**68.3**	67.8	78.3	**59.7**	76.9	**58.5**	**63.34**	59.5	**69.87**
Improvement	-	-	**0.88**	**10.5**	**3.92**	-	-	**2.48**	-	**2.6**	**5.04**	-	**3.17**

**Table 6 sensors-24-07215-t006:** Gaussian noise addition comparison of various 3D segmentation models. The bold data shows the best performing methods while the underlined ones indicate the second-best performing methods.

Model	mAP@50%	mAP@50% (When Noise Added)	Degradation
3D-BoNet [39]	66.7	61.3	5.4
3D-BEVIS [44]	65.6	60.5	**5.1**
SGPN [41]	42.9	35.6	7.3
3D ANSS GHT (Ours)	**69.87**	66.74	**2.13**

**Table 7 sensors-24-07215-t007:** Computation speed comparison of n-sigmoid and n-shifted sigmoid activation functions on the 3D ANSS GHT and other 3D models using S3DIS dataset using S3DIS dataset.

Model	Activation Function	Inference Time (ms)
3D ANSS GHT	n-Sigmoid	29.35
	n-shifted Sigmoid	**28.17**
3D-MPA [40]	-	300
SGPN [41]	-	170
DetectionNet [45]	-	739

**Table 8 sensors-24-07215-t008:** Effects of other activation functions on the 3D ANSS GHT using the S3DIS dataset. The bold data shows the best performing methods while the underlined ones indicate the second-best performing methods.

Model	Activation Function	mAP@50%
3D ANSS GHT	Original Sigmoid	41.3
	ReLU	43.4
	ELU	54.17
	n-Sigmoid	63.35
	n-Shifted Sigmoid	**69.87**

**Table 9 sensors-24-07215-t009:** Effects of attention mechanism on the 3D ANSS GHT using the S3DIS dataset. The bold data shows the best performing methods while the underlined ones indicate the second-best performing methods.

Model	Attention Mechanism	mAP@50%
3D ANSS GHT	No Attention	62.53
	ANSS	**69.87**
	Traditional Shuffle	66.29

## Data Availability

The dataset used for this study is openly available https://rgbd.cs.princeton.edu (accessed on 6 November 2024).

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
