# Peer review of "Three-Dimensional Instance Segmentation Using the Generalized Hough Transform and the Adaptive n-Shifted Shuffle Attention"

_sensors, 2024, doi:10.3390/s24227215_

Round 1

Reviewer 1 Report

Comments and Suggestions for Authors

This paper introduces a n-shifted Sigmoid Channel and Spatial Attention Module for Efficient 3D Scene Object Detection. However, the paper’s flawed writing format and dearth of detail render it unreadable, and it is further hindered by a lack of innovation. The detailed comments are listed as follows:

1) Although the paper touts the n-shifted sigmoid CSA layer as its primary contribution, its reliance on the previously published n-shifted sigmoid activation function and CSA modules lacks novelty, as it simply combines existing components without introducing original concepts.

2) Figures 1 through 5, which currently appear too rudimentary, require redrawing with enhanced detail to improve clarity and comprehension.   

3) The discussion of related works is incomplete, for example the published n-shifted sigmoid activation has not been discussed.

4) The content of the methodology can be further refined and focused, and redundant content should be reduced.

5) Numerous formatting issues require correction, including the excessive indentation on line 44 and the incorrect formatting of several tables.

6) The Python code would be better suited for hosting on a platform like GitHub, rather than included directly within the paper.

Comments on the Quality of English Language

The English language quality is generally acceptable, though some moderate editing is needed for improvement.

Author Response

This paper introduces a n-shifted Sigmoid Channel and Spatial Attention Module for Efficient 3D Scene Object Detection. However, the paper’s flawed writing format and dearth of detail render it unreadable, and it is further hindered by a lack of innovation. The detailed comments are listed as follows:

1) Although the paper touts the n-shifted sigmoid CSA layer as its primary contribution, its reliance on the previously published n-shifted sigmoid activation function and CSA modules lacks novelty, as it simply combines existing components without introducing original concepts.

The authors have decided to completely overhaul the paper to bring in more innovations such that the here:

  • the n-shifted sigmoid is an evolution from the n-sigmoid which is designed to address its limitations by shifting the output range to [-1, 1]. Unlike the n-sigmoid, which outputs values between [0,1], this shifted version is symmetrical around zero, producing values in the range [−1,1]. This symmetry can improve gradient flow and model convergence by allowing both positive and negative responses to the input data.
  • The CSA module has been removed completely to be replaced by aninnovative adaptive shuffle attention module which the adaptive part of this attention layer ensures it dynamically adjusts the shuffle pattern, allowing it to capture relationships between features more effectively over time. By learning to shuffle feature maps, this method enhances information flow and improves the model’s ability to detect objects in varying orientations and scales.

2) Figures 1 through 5, which currently appear too rudimentary, require redrawing with enhanced detail to improve clarity and comprehension.   Figure 1 changed completely

3) The discussion of related works is incomplete, for example the published n-shifted sigmoid activation has not been discussed. The authors have added the n-shifted sigmoid activation in the related works

4) The content of the methodology can be further refined and focused, and redundant content should be reduced. The authors have have redefined the methodology

5) Numerous formatting issues require correction, including the excessive indentation on line 44 and the incorrect formatting of several tables. The authors have fixed the indentation problems.

6) The Python code would be better suited for hosting on a platform like GitHub, rather than included directly within the paper. The python code has been removed.

Reviewer 2 Report

Comments and Suggestions for Authors

In this manuscript, An Innovative n-shifted Sigmoid Channel and Spatial Attention Module for Efficient 3D Scene Object Detection is proposed, and the authors consider using 3D scene object detection in the deep Hough voting point sets as a testing application. However, there are still some issues worth mentioning.

1. Combining the deep Hough voting model with channel and spatial attention models lacks sufficient innovativeness. It would be better to incorporate additional innovative elements if possible. 2. You should give a further introduction to the n-shifted sigmoid channel and spatial attention model.

3. There are many problems with the layout of tables, images (such as Table 1 and Figure 4.) and the formatting of the first line of paragraphs in this article, and you must pay attention to the formatting in the process of writing the article.

        4. In this manuscript, there is a lack of sufficient experimental data to demonstrate the feasibility of 3D object detection, and you can add a comparison experiment to test the effects of n-sigmoid channel and spatial attention and the combination of the two modules separately. You can conduct experiments on more datasets to verify the generalization and applicability of object detection in 3D scenes in this paper.

        5. For the code section, please use the pseudo-code format for the demonstration.

Comments on the Quality of English Language

The English language in this manuscript is of good quality and there are no comments.

Author Response

In this manuscript, An Innovative n-shifted Sigmoid Channel and Spatial Attention Module for Efficient 3D Scene Object Detection is proposed, and the authors consider using 3D scene object detection in the deep Hough voting point sets as a testing application. However, there are still some issues worth mentioning.

Combining the deep Hough voting model with channel and spatial attention models lacks sufficient innovativeness. It would be better to incorporate additional innovative elements if possible.  The authors have brought innovations with the adaptive n-shifted shuffle attention which comprise the n-shifted sigmoid activation. This is an evolution from the n-sigmoid activation function [7] which is designed to address its limitations by shifting the output range to [-1, 1]. Unlike the n-sigmoid, which outputs values between [0,1], this shifted version is symmetrical around zero, producing values in the range [−1,1]. This symmetry can improve gradient flow and model convergence by allowing both positive and negative responses to the input data but also the adaptive shuffle attention which dynamically adjusts the shuffle pattern, allowing it to capture relationships between features more effectively over time. By learning to shuffle feature maps, this method enhances information flow and improves the model’s ability to detect objects in varying orientations and scales.

  1. You should give a further introduction to the n-shifted sigmoid channel and spatial attention model.

Section 3.3 introduces the innovative Adaptive n-shifted Shuffle Attention Mechanism

  1. There are many problems with the layout of tables, images (such as Table 1 and Figure 4.) and the formatting of the first line of paragraphs in this article, and you must pay attention to the formatting in the process of writing the article.

The authors have fixed the formatting issues.

  1. In this manuscript, there is a lack of sufficient experimental data to demonstrate the feasibility of 3D object detection, and you can add a comparison experiment to test the effects of n-sigmoid channel and spatial attention and the combination of the two modules separately. You can conduct experiments on more datasets to verify the generalization and applicability of object detection in 3D scenes in this paper.

 Extensive experiments and ablation studies have been conducted to validate the proposed model.

  1. For the code section, please use the pseudo-code format for the demonstration.

The code has been removed and will be in a near future to the Github website.

Reviewer 3 Report

Comments and Suggestions for Authors

 The authors present a  effective approach to 3D object detection using a new n-shifted sigmoid channel and spatial attention module. This innovative method significantly improves detection accuracy, as evidenced by comprehensive empirical results. The research design is robust, and the conclusions are well-supported by the data.

However, to enhance the manuscript's clarity and readability, some minor revisions are necessary. Specifically, the methods section would benefit from additional explanations and visual aids to clarify complex concepts. Furthermore, minor grammatical errors and awkward phrasing should be corrected to improve the overall quality of the text.

Author Response

 The authors present a  effective approach to 3D object detection using a new n-shifted sigmoid channel and spatial attention module. This innovative method significantly improves detection accuracy, as evidenced by comprehensive empirical results. The research design is robust, and the conclusions are well-supported by the data.

However, to enhance the manuscript's clarity and readability, some minor revisions are necessary.

The authors have completely revised the paper introducing new concepts such as the adaptive shuffle attention and an evolution from the published n-sigmoid activation function.

Specifically, the methods section would benefit from additional explanations and visual aids to clarify complex concepts.

The method section has been completely rewritten.

Furthermore, minor grammatical errors and awkward phrasing should be corrected to improve the overall quality of the text.

The authors have fixed the grammatical errors and awkward phrasing.

Round 2

Reviewer 1 Report

Comments and Suggestions for Authors

This paper introduce a novel approach which combines ANSS attention mechanisms and GHT for 3D instance segmentation. However, the lack of details in the proposed method renders the paper challenging to comprehend. The detailed comments are as follows:

1.     The paper declares that the GHT has been reformulated for integration into a deep learning framework, yet it fails to provide detailed descriptions of the reformulation and integration.

2.     Fig 1 requires redrafting as the current version lacks sufficient detail and is not informative.

3.     The paper lists several benefits of the n-shifted sigmoid function within attention mechanisms, yet it lacks explanations and derivations to support these claims.

4.     Line 80, the performance improvement on the S3DIS dataset should not be considered a distinct contribution, as it is merely an outcome of the introduced modules.

Comments on the Quality of English Language

The entire manuscript requires thorough revision to correct typos and to improve the English writing. 

Author Response

The authors would like to express gratitude for the opportunity given to make changes to the paper and for the insightful suggestions.

First, the authors have addressed the following preoccupations:

Is the research design appropriate?

The authors have decided to add a new figure (Figure 2) which clearly describes in detail the operations in the ANSS.

Are methods adequately described?

The authors have completely modified section 3 (The proposed Adaptive n-shifted Shuffle (ANSS) Attention Integrated with Generalized Hough Transform (GHT)) to make it clearer. Since the proposed model has two main parts, section 3 was also divided in 2 sub-sections (ANSS and GHT). The ANSS was now also divided into its 3 parts which are the ? (n-shifted sigmoid) section 3.1.1., AS (Adaptive Shuffle) section 3.1.2. and SA (Shuffle Attention) section 3.1.3. The GHT was also divided into the PointNet backbone (section 3.2.1.) and the GHT module (section 3.2.2.) which explains its evolution to a deep learning framework for 3D object detection.

Are the results clearly presented?

The authors have decided to redesign the results section completely. They have now divided it into:

  1. Dataset and Evaluation metrics with Data preprocessing and implementation details sub-sections.
  2. Results and Analysis where the performance of the proposed method on AR, Ap, robustness to noise and inference time are presented and discussed.
  3. Ablation studies where the various impact of the n-shifted sigmoid, the ANSS and the GHT are discussed.
  4. Visualization of results for qualitative results and failure cases.
  5. Discussion which also contains the future research directions

Are the conclusions supported by the results?

The conclusions were completely rewritten so that they will define once again the proposed methodology in few words, show its added advantages over existing state-of-the-art methods and confirm that the experiments conducted have demonstrated this superiority.

Comments and Suggestions for Authors

This paper introduced a novel approach which combines ANSS attention mechanisms and GHT for 3D instance segmentation. However, the lack of details in the proposed method renders the paper challenging to comprehend. The detailed comments are as follows:

  1. The paper declares that the GHT has been reformulated for integration into a deep learning framework, yet it fails to provide detailed descriptions of the reformulation and integration.

This reformulation involves modifications to make the process differentiable, scalable, and trainable within a neural network structure. The GHT can be modified to work with continuous probability density functions, so the vote accumulation in the Hough space becomes a continuous, smooth function. This is achieved by differentiable functions such as softmax, sigmoid, n-shifted sigmoid …

Here the network outputs these representations (often termed as "voting vectors") that encode position, scale, orientation, and other parameters. Here in the ANSS GHT, the n-shifted sigmoid function definitely allows to scale the voting intensities based on learned parameters, dynamically adapting to various spatial patterns and contexts in the 3D space.

The reformulated GHT benefits from convolutional layers to extract hierarchical features from the input (e.g., 3D points clouds). By using a convolutional backbone, the network can efficiently capture both local and global context.

A thorough explanation has been added in section 3.2 starting from 3.2.1.

  1. Fig 1 requires redrafting as the current version lacks sufficient detail and is not informative. More information added to the figure where the three parts of The ANSS (Adaptive Shuffle AS, n-shifted Sigmoid activation function ?, and Shuffle Attention SA) are explained in more details.

A Figure 2 has been added to better showcase the various components involved in the 3 parts. The Adaptive Shuffle (AS) with conv1x1, then channels are shuffled according to a learned shuffle pattern. The n-shifted activation function adds flexibility in scaling and focusing. It introduces a learnable shift parameter that shifts the activation range of the sigmoid for finer control over the activation output to adjust the focus area. The Shuffle Attention redistribute dynamically the attention weights across feature channels. Its layer comprises a Channel-wise Pooling to generate a channel-wise attention vector. Then an attention weight calculation where the pooled outputs pass through a pair of fully connected layers with a ReLU activation in between before the n-shifted sigmoid. Finally, the attention weight A is applied to the feature map X' to produce the attended feature map.

A thorough explanation of the ANSS 3 parts is given in Section 3 just beneath the figure where mathematical formulas, layer explanation… are provided. Figure 2 is proposed to explain the various operations taking place in the ANSS.

  1. The paper lists several benefits of the n-shifted sigmoid function within attention mechanisms, yet it lacks explanations and derivations to support these claims.

  • Enhanced Representation of Negative Relationships:

To shift this range to [−1,1], the authors introduce the  parameter, and rescale the output:

                                                            ?                                   

With this adjustment, ? now outputs values in the range [−1,1], allowing the model to recognize negative relationships where attention needs to be reduced or negated.

-             Improved Attention Weights

The broadened output range of [−1,1] enables finer-grained control over attention weights:

  • values near 1 represent strong positive influence, amplifying attention toward crucial features.
  • values near −1 indicate strong negative influence, which suppresses or disregards elements that could hinder the model’s performance.

The weighted attention A for a feature f is calculated by applying ? ​:

                                                                      A(f) = f · ?       

-          Enhanced Model Stability and Training            

With n-shifted Sigmoid, this gradient issue is reduced, mainly because:

  1. The output range [−1,1] allows for more substantial gradients even for large inputs.
  2. The shift parameter  can be adjusted during training, adapting the activation to changing input distributions.

The broader range ensures that even for extreme values of , the gradient does not diminish to zero as quickly as in the standard sigmoid function, preserving gradient flow.

-                    Adaptability to Complex Tasks

For a feature map  with boundary between regions R1 and R2​, the attention weights (Equation 10) from ?  will enhance FR1 while attenuating FR2 ​​, yielding:

                                                          A() = [?  ≈ 1, ?  ≈ -1 ]                                 (10)

This adaptability allows the model to effectively learn complex spatial relationships, leading to more accurate segmentation and classification in 3D scenarios.

In section 3.4.3. where benefits of the n-shifted sigmoid were listed, detailed explanation with mathematical formulas are provided to support them.

  1. Line 80, the performance improvement on the S3DIS dataset should not be considered a distinct contribution, as it is merely an outcome of the introduced modules.

The authors would merely like to emphasize that the performance improvement on the S3DIS dataset is just a reflection of the robustness of the proposed model with the combination of the ANSS and the GHT.

Comments on the Quality of English Language

The entire manuscript requires thorough revision to correct typos and to improve the English writing. 

The entire manuscript has been revised and recomposed to address the reviewer's suggestions and improve the writing.

Submission Date

27 May 2024

Date of this review

15 Oct 2024 18:44:43

Reviewer 2 Report

Comments and Suggestions for Authors

### Review Comments Translation

1. The format of each table should be more uniform, and for Table 4, data for the same network could be placed in the same row for comparison to enhance clarity.

2. The formatting of equations, such as Equation 1 and Equation 2, needs improvement, including centering, font size, and spacing between characters.

3. Grammar mistakes consistently occur throughout the paper, for example in line 82: "The rest of this paper is is organized as follows:". Please thoroughly review these issues.

4. The diagram illustrating the network is oversimplified. ANSS and GHT structures should be shown more in detail.

5. In the objective analysis of the paper, the author does not fully explain the meaning of a single indicator, which makes it weakly relate to the subjective evaluation.

Comments on the Quality of English Language

The English language in this manuscript is of good quality and there are no comments.

Author Response

The authors would like to express gratitude for the opportunity given to make changes to the paper and for the insightful suggestions.

First, the authors have addressed the following preoccupations:

Is the research design appropriate?

The authors have decided to add a new figure (Figure 2) which clearly describes in detail the operations in the ANSS.

Are methods adequately described?

The authors have completely modified section 3 (The proposed Adaptive n-shifted Shuffle (ANSS) Attention Integrated with Generalized Hough Transform (GHT)) to make it clearer. Since the proposed model has two main parts, section 3 was also divided in 2 sub-sections (ANSS and GHT). The ANSS was now also divided into its 3 parts which are the ? (n-shifted sigmoid) section 3.1.1., AS (Adaptive Shuffle) section 3.1.2. and SA (Shuffle Attention) section 3.1.3. The GHT was also divided into the PointNet backbone (section 3.2.1.) and the GHT module (section 3.2.2.) which explains its evolution to a deep learning framework for 3D object detection.

Are the results clearly presented?

The authors have decided to redesign the results section completely. They have now divided it into:

  1. Dataset and Evaluation metrics with Data preprocessing and implementation details sub-sections.
  2. Results and Analysis where the performance of the proposed method on AR, Ap, robustness to noise and inference time are presented and discussed.
  3. Ablation studies where the various impact of the n-shifted sigmoid, the ANSS and the GHT are discussed.
  4. Visualization of results for qualitative results and failure cases.
  5. Discussion which also contains the future research directions

Are the conclusions supported by the results?

The conclusions were completely rewritten so that they will define the proposed methodology in few words, show it added advantages over existing state-of-the-art methods and confirm that the experiments conducted have demonstrated this superiority.

Comments and Suggestions for Authors

### Review Comments Translation

  1. The format of each table should be more uniform, and for Table 4, data for the same network could be placed in the same row for comparison to enhance clarity. The authors have fixed the format of all tables to make them uniform. Some tables are bigger so the authors have made them in the center of the whole page. The have resolve the issue of lack of clarity for results comparison and made the results on the same row (Table 3 and 4) as suggested.
  2. The formatting of equations, such as Equation 1 and Equation 2, needs improvement, including centering, font size, and spacing between characters. The authors have centered and numbered all equations.
  3. Grammar mistakes consistently occur throughout the paper, for example in line 82: "The rest of this paper is is organized as follows:". Please thoroughly review these issues. The paper has been recomposed to align better with the reviewers suggestions and the mdpi best presentation practices. The authors have re-read and recomposed the whole paper to fix grammatical issues and address the reviewer's suggestions..
  4. The diagram illustrating the network is oversimplified. ANSS and GHT structures should be shown more in detail. The authors have added names for the 3 parts of the ANSS in Figure 1. They have chosen to add  an extra figure (Figure 2) for more explanation where the various steps of the proposed methodology have been shown according to the part they belong to.
  5. In the objective analysis of the paper, the author does not fully explain the meaning of a single indicator, which makes it weakly relate to the subjective evaluation.

In 3D instance segmentation, mean average recall (mAR), mean average precision (mAP), robustness to noise, inference time… constate the most reliable indicators of the validity of a methodology. The authors have added a brief explanation of these critical indicators at the beginning of each results presentation.

Comments on the Quality of English Language

The English language in this manuscript is of good quality and there are no comments.

Submission Date

27 May 2024

Date of this review

23 Oct 2024 09:45:34